# Multi-Modality Co-Learning for Efficient Skeleton-based Action Recognition

## ABSTRACT

Skeleton-based action recognition has garnered significant attention due to the utilization of concise and resilient skeletons. Nevertheless, the absence of detailed body information in skeletons restricts performance, while other multimodal methods require substantial inference resources and are inefficient when using multimodal data during both training and inference stages. To address this and fully harness the complementary multimodal features, we propose a novel multi-modality co-learning (MMCL) framework by leveraging the multimodal large language models (LLMs) as auxiliary networks for efficient skeleton-based action recognition, which engages in multi-modality co-learning during the training stage and keeps efficiency by employing only concise skeletons in inference. Our MMCL framework primarily consists of two modules. First, the Feature Alignment Module (FAM) extracts rich RGB features from video frames and aligns them with global skeleton features via contrastive learning. Second, the Feature Refinement Module (FRM) uses RGB images with temporal information and text instruction to generate instructive features based on the powerful generalization of multimodal LLMs. These instructive text features will further refine the classification scores and the refined scores will enhance the model's robustness and generalization in a manner similar to soft labels. Extensive experiments on NTU RGB+D, NTU RGB+D 120 and Northwestern-UCLA benchmarks consistently verify the effectiveness of our MMCL, which outperforms the existing skeleton-based action recognition methods. Meanwhile, experiments on UTD-MHAD and SYSU-Action datasets demonstrate the commendable generalization of our MMCL in zero-shot and domain-adaptive action recognition. Our code will be publicly available and can be found in the supplementary files.

## CCS CONCEPTS

• **Computing methodologies → Activity recognition and understanding**.

## KEYWORDS

Action Recognition, Multi-modality, Multimodal LLMs

**ACM Reference Format:**
. 2024. Multi-Modality Co-Learning for Efficient Skeleton-based Action Recognition. In *Proceedings of 32st ACM International Conference on Multimedia (MM '24)*. ACM, New York, NY, USA, 10 pages. https://doi.org/XXXXXXX.XXXXXXX

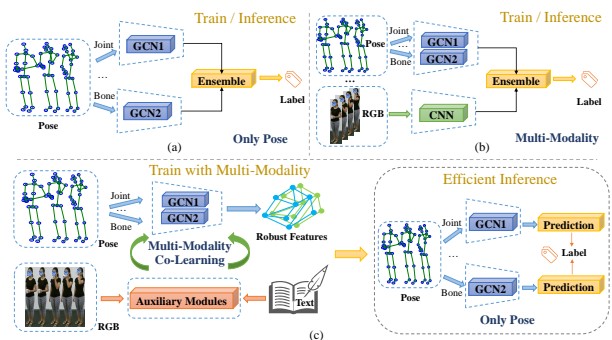

**Figure 1: Existing methods suffer from inherent defects in single-modality and issues of inefficient inference. (a) Most skeleton-based methods only use skeleton/pose modality during training and inference stages, encountering issues associated with skeletal inherent defects. Note that the human pose can be divided into different skeleton modalities (e.g. joint and bone). (b) Most multimodal-based methods use multi-modality during the training and inference stages, which require significant inference resources and are inefficient. (c) Our multi-modality co-learning (MMCL) framework incorporates multimodal features to enhance the modeling of skeletons in the training stage and maintains efficiency in the inference stage by only using concise skeletons.**

## 1 INTRODUCTION

Human action recognition is an important task in video understanding. The action conveys central information like body tendencies and thus helps to understand the person in videos. In pursuit of precise action recognition, diverse video modalities have been explored, such as skeleton sequences [6, 19], RGB images [13, 61], text descriptions [49, 56] and depth images [54]. In particular, the graph-structured skeleton can well represent body movements and is highly robust to environmental changes, thereby adopted in many studies using graph convolutional networks (GCNs) [7–9, 12, 30, 36, 50], it also demonstrates strong scalability and real-time capabilities when deployed on edge devices. However, these skeleton-based methods [19, 39, 53, 65] (Fig. 1 (a)) only using skeletons during training and inference stages will restrict recognition performance due to the inherent defects of skeletal modality. For instance, the skeleton modality lacks the ability to depict detailed body information (e.g. appearance and objects) and has difficulty in fine-grained recognition when dealing with similar actions.

A good approach to addressing the aforementioned skeleton-based limitations is to introduce complementary multi-modality for action recognition. The typical process [1, 10, 13, 23, 61] involves employing two networks to separately model the skeletons and other modalities (e.g. RGB images and depth images), then some ensemble techniques are applied to merge the predictions from

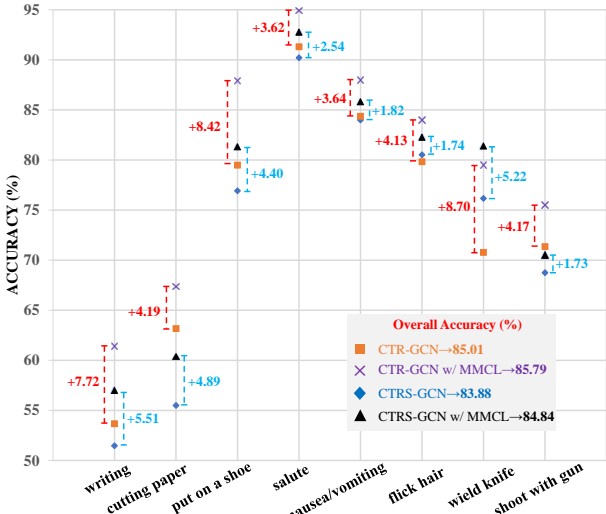

**Figure 2: Comparisons of different backbones in recognition performance. We apply the MMCL to GCN backbones and observe its capability to aid in better modeling. Specifically, we show the overall accuracy using joint modality on the NTU120 XSub benchmark and the top-8 actions with the highest improvement.**

multiple modalities. These multimodal-based methods (Fig. 1 (b)) can make up for the defects of a single modality and provide more comprehensive information for fine-grained action classification by leveraging the complementary nature of multimodal data. Nevertheless, they suffer from the drawbacks of requiring significant inference resources and appear less efficient when deployed on edge devices due to the use of multi-modality in both training and inference stages(Fig. 1 (b)). The aforementioned skeleton-based and multimodal-based methods naturally lead to a question: *How to better leverage the complementary nature of multi-modality while retaining the efficient inference with single-skeleton modality?*

Motivated by the above question, we propose a multi-modality co-learning (MMCL) framework for efficient skeleton-based action recognition, which enhances the model's performance and generalization via multi-modality co-learning, while only using the concise skeleton in the inference stage to preserve the efficiency (Fig. 1 (c)). Due to the effectiveness of multi-modality co-learning during the training stage, our proposed MMCL acquires enhanced robustness and generalization when applied to optimize mainstream GCN backbones, exhibiting improved performance in overall accuracy and specific actions as shown in Fig. 2.

In our MMCL framework, we incorporate the multi-modality co-learning into action recognition and provide instructive multimodal features based on the multimodal LLMs [14, 27, 28, 66] for skeletons. The proposed MMCL is shown in Fig. 3, which will use skeleton, RGB and text modalities during the training stage, while only using the skeleton in inference to achieve efficiency. Given that RGB features [10, 61] effectively compensate for the skeletons, while existing multimodal LLMs exhibit strong generalization on real RGB images and achieve success in various visual

tasks [15, 63, 67], we adopt the RGB images as the input for the multimodal LLMs to help better model the skeletons. To mitigate the limitation of some existing multimodal LLMs (e.g. BLIP [28], BLIP2 [27], MiniGPT-4 [66] and LLaMA AdapterV2 [14]) that cannot directly process video streams, we combine frames from the video stream into RGB images that include temporal information. Besides, to better harness the potent modeling capabilities of these multimodal LLMs for intrinsic features in images and text, we input the RGB images with text instructions into the Feature Refinement Module (FRM) for generating instructive text features conducive to action recognition. These instructive text features from multimodal LLMs will further refine the classification scores output by the fully connected layer and enhance the model's robustness and generalization in a manner similar to soft labels. Meanwhile, to better utilize the introduced RGB images with temporal information in LLMs, our MMCL uses a deep neural network to obtain RGB features in the Feature Alignment Module (FAM). Likewise, inspired by CLIP [42], our MMCL uses the contrastive learning [56, 64] to better assist in modeling skeletons.

Our contributions are summarized as follows:
- We propose a novel multi-modality co-learning (MMCL) framework for efficient skeleton-based action recognition, which empowers mainstream GCN models to produce more robust and generalized feature representations by introducing multi-modality co-learning during the training stage, while maintain efficiency by only using concise skeletons in inference.
- Our proposed MMCL framework is the first to introduce multimodal LLMs for multi-modality co-learning in skeleton-based action recognition. Meanwhile, our MMCL is orthogonal to the backbones and thus can be applied to optimize mainstream GCN models by using different multimodal LLMs. Due to the generalization of multimodal LLMs, our MMCL can be transferred to domain-adaptive and zero-shot action recognition.
- Extensive experiments on three popular benchmarks namely NTU RGB+D, NTU RGB+D 120 and Northwestern-UCLA datasets verify the effect of our MMCL framework by outperforming existing skeleton-based methods. Meanwhile, experiments on SYSU-ACTION and UTD-MHAD datasets from different domains indicate that our MMCL exhibits commendable generalization in both domain adaptive and zero-shot action recognition.

## 2 RELATED WORK

### 2.1 Skeleton-based Action Recognition

Deep learning-based methods[37, 57, 59] have achieved high success using CNNs [21, 22, 26, 32, 55] and RNNs [2, 25, 29, 35, 40, 44, 47, 62] in skeleton-based action recognition. To address the adverse effects due to view variations and noisy data, Liu et al. [37] visualized skeleton sequences as color images and fed them into a multi-stream CNN for deep feature extraction. Wang et al. [47] proposed a two-stream RNN to model the temporal dynamics and spatial structure of skeleton sequences, which breaks through the limitations of RNN in processing raw skeleton data. GCNs can effectively process structured data, thereby being adopted by many researchers in skeleton-based action recognition. For instance, Yan et al. [59] is the first to use GCNs for skeleton-based action recognition by proposing the ST-GCN to learn spatiotemporal features

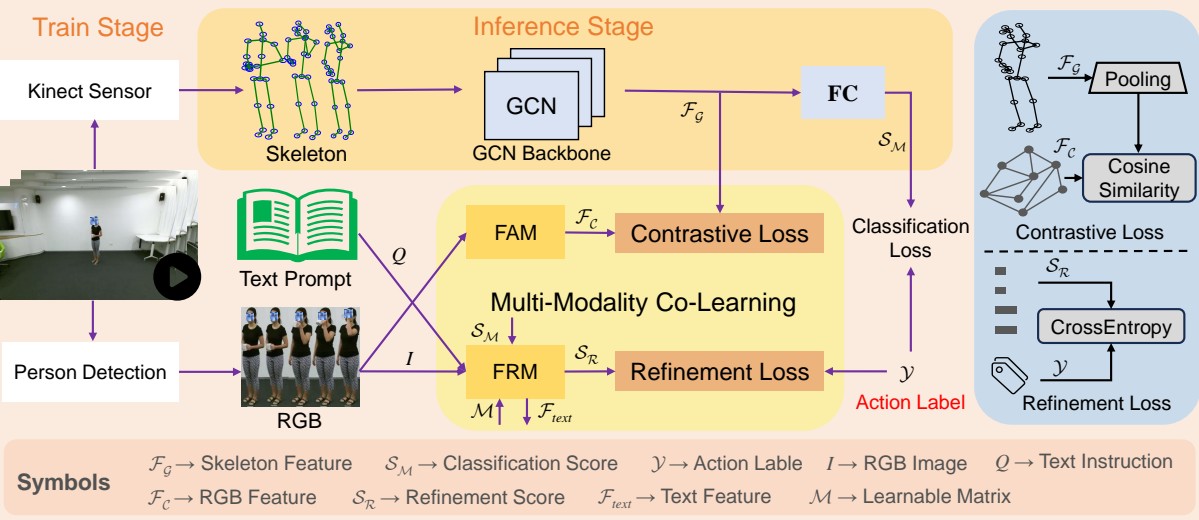

**Figure 3: Framework of our proposed Multi-Modality Co-Learning (MMCL), which integrates multimodal features during the training stage and keeps efficiency in inference by only using concise skeletons. The Feature Alignment Module (FAM) extracts and aligns high-level RGB features to facilitate contrastive learning with global skeleton features. Here, we only align the RGB features with the skeleton features as the text features generated by LLMs have relatively limited information compared to the RGB features. The Feature Refinement Module (FRM) provides instructive text features to refine the classification scores based on the multimodal LLMs. Here we guide the LLMs to generate instructive features based on the defects that the skeleton cannot recognize objects. The text instructions can be modified based on the skeletal defects.**

from skeleton data. Chen et al. [6] improve the design of GCNs by proposing a channel-wise topology refinement graph convolutional network (CTR-GCN), which effectively aggregates joint features in each channel. The methods mentioned above are limited by the drawbacks of only using skeleton modality.

## 2.2 Multimodal-based Action Recognition

Benefiting from the emergence of various multimodal datasets and improvements in computing resources, research about multimodal-based action recognition [1, 10, 13, 23, 61] has become popular. To address GCNs are subject to limitations in robustness, interoperability and scalability, Duan et al. [13] proposed a novel PoseConv3D to use both RGB heatmap and skeleton modalities for robust human action recognition. Das et al. [10] proposed a video-pose network (VPN) to project the 3D poses and RGB cues in a common semantic space, which enabled the action recognition framework to learn better spatiotemporal features exploiting both modalities. Commonly used modalities for multimodal-based action recognition include skeletons, color images, depth maps, text and point clouds. Methods [10, 11, 13, 20, 41, 54, 61] of integrating these multiple modalities have been shown to achieve better performance by leveraging multimodal features. Unlike the above methods that use multimodal data in both training and inference stages, our MMCL introduces multimodal data for multi-modality co-learning in the training stage and only uses the concise skeleton to keep efficient in inference.

## 2.3 Large Language Model Auxiliary Learning

The powerful generation ability of large language models (LLMs) enables them to be transferred to different tasks effectively. Prompt

learning and instruction learning are capable techniques of adapting different pre-trained LLMs to different tasks, thereby being applied in action recognition as an auxiliary strategy by many researchers. Wang et al. [49] proposed an ActionCLIP to directly uses class labels as input text to construct cues for video action recognition. Xiang et al. [56] used LLMs based on action labels to provide prior knowledge for skeleton-based action recognition. Different from the above methods that use simple text modalities based on action labels and unimodal LLMs, our MMCL is based on advanced multimodal LLMs for multi-modality co-learning and sets the text instructions according to the skeletal defects. Besides, these methods [49, 56, 58] that explicitly generate text features based on action labels are not advantageous for unknown actions. With the emergence of multimodal LLMs such as BLIP [28], BLIP2 [27], MiniGPT-4 [66] and LLaMA AdapterV2 [14], receiving multimodal input can often generate more robust and richer text information. Inspired by this, our MMCL framework inputs RGB and text modality into multimodal LLMs and obtains the output text features for refining the classification score of the skeleton. These refined scores will enhance the model's robustness and generalization in a manner similar to soft labels. Benefiting from the powerful generalization capabilities of multimodal LLMs, the soft labels obtained by MMCL can be transferred to domain-adaptive action recognition.

## 3 METHODOLOGY

In this section, we present our proposed Multi-Modality Co-Learning (MMCL) framework in detail. MMCL aims to enhance skeleton

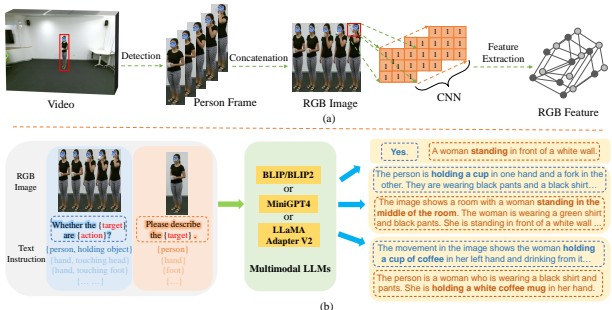

**Figure 4: (a) Extract RGB images from video and use CNN to model RGB features. (b) Display of content generated by different multimodal LLMs. The text instructions used by MMCL are set based on skeletal defects (e.g. lack of object information and appearance details), guiding the LLMs to generate features complementary to the skeleton from RGB images. In implementation, we just use the BLIP [28] to generate text features for training.**

representation learning with complementary and instructive multimodal features based on the powerful generalization capabilities of multimodal LLMs. Meanwhile, MMCL is orthogonal to the backbone networks and thus can be coupled with various GCN backbones and multimodal large language models. Below we will introduce the proposed MMCL in five parts.

## 3.1 GCN Backbone

Due to the unique advantages of GCNs in modeling graph-structured data, the GCN is prevailing for skeleton-based action recognition. In our MMCL framework, we also adopt GCN as the backbone network to model the skeleton modality and the GCN blocks are composed of a graph convolution layer and a temporal convolution layer. The normal graph convolution can be formulated as:

$$H^{l+1} = \sigma \left( D^{-\frac{1}{2}} A D^{-\frac{1}{2}} H^l W^l \right), \tag{1}$$

where $H^l$ is the joint features at layer $l$ and $\sigma$ is the activation function. $D \in R^{N \times N}$ is the degree matrix of $N$ joints and $W^l$ is the learnable parameter of the $l$-th layer. $A$ is the adjacency matrix representing joint connections. Generally, the $A$ can be generated by using static and dynamic ways. The $A$ is generated by using data-driven strategies in dynamic ways while it is defined manually in static ways. [6] introduces $A$ into graph convolution in the form of $S = (s_{id}, s_{cp}, s_{cf})$ and $s_{id}, s_{cp}, s_{cf}$ indicate the identity, centripetal and centrifugal edge subsets respectively. [24] proposes an HD-Graph to introduce $A$ into graph convolution and uses a six-way ensemble strategy to train the GCN.

## 3.2 Multimodal Data Processing

In our MMCL framework, we use skeleton modality, RGB modality and text modality during the training stage, while only using the concise skeleton modality during the inference stage. The skeleton data is collected through Kinect sensors and it is a natural topological graph, which is denoted as $G = \{V, E\}$. $V = \{v_1, v_2, \cdots, v_N\}$

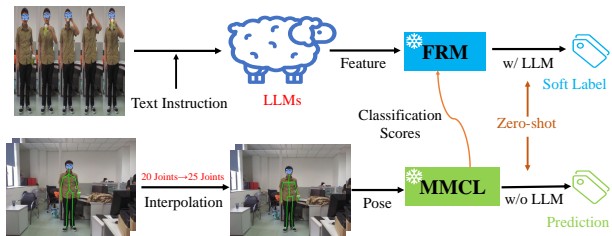

**Figure 5: Our MMCL can effectively perform action recognition when faced with skeleton/pose inputs from different domains. Our MMCL employs skeleton interpolation to ensure that the number of skeleton points input to the model is consistent.**

and $E = \{e_1, e_2, \cdots, e_L\}$ represent $N$ joints and $L$ bones. Our MMCL uses four different skeleton modalities, namely joint, bone, joint motion and bone motion defined in [36].

Since the RGB modality has rich human body information, we extract three-channel RGB images from videos to assist the learning of skeletons based on multimodal LLMs. In Fig. 4(a), considering that most existing LLMs are more proficient in handling individual image-text pairs, we combine video frames into an RGB image with temporal information. For example, we select $m$ frames from a action video of drinking water and concatenate them from left to right in chronological order, thus forming an RGB image containing the temporal information of drinking water, which is show in figure 4(a). In detail, given an RGB video stream $K = \{k_1, k_2, \cdots, k_n\}$ with $n$ frames, we extract the three-channel RGB image $I$ by:

$$I = \oplus_m U_n^m (\Gamma(\varepsilon(k_1)), \Gamma(\varepsilon(k_2)), \cdots, \Gamma(\varepsilon(k_n))), \tag{2}$$

where $\varepsilon$ denotes a detector. Inspired by the top-down pose estimation, we use a person detector $\varepsilon$ to filter out most ambient noise and focus on the human body. Meanwhile, we resize the extracted person frame to save computing and storage resources by $\Gamma$. This detection and cropping method significantly filters environmental noise, while minimizing the loss of spatial information and has been widely applied in [33, 61]. Inspired by the uniform sampling strategy in [13], we uniformly sample $m$ samples from $n$ person frames to reduce the temporal dimension by $U_n^m$. $\oplus_m = [f_1 \oplus f_2 \oplus \cdots \oplus f_{m-1} \oplus f_m]$ means to concat $m$ person frames along the temporal dimension to form the RGB image with temporal information.

## 3.3 Multi-Modality Co-Learning

*3.3.1 Feature Alignment Module.* In our MMCL framework, the Feature Alignment Module (FAM) extracts rich RGB features and aligns them with global skeleton features from GCN layers by:

$$\mathcal{F}_c = \mathcal{M}_{align}(\Theta(Norm(\Gamma(I)))). \tag{3}$$

Due to the CNNs have unique advantages in processing Euclidean data (e.g. RGB images), the FAM utilizes a deep convolutional neural network $\Theta$ to model high-level features from RGB image $I$. Before extracting RGB features, the FAM will resize and normalize the image to speed up the convergence of the model. In order to achieve

**Table 1: Accuracy improvement of some action categories when MMCL assists GCN backbones.**

| Action Label | 2 | 4 | 12 | 16 | 23 | 24 | 28 | 31 | 33 |
|---|---|---|---|---|---|---|---|---|---|
| Acc. (CTRS-GCN) (%) | 68.73 | 86.81 | 51.47 | 76.92 | 90.15 | 94.20 | 81.82 | 77.90 | 86.23 |
| Acc. (CTRS-GCN w/ MMCL) (%) | $70.55^{\uparrow 1.82}$ | $89.01^{\uparrow 2.20}$ | $56.99^{\uparrow 5.52}$ | $81.32^{\uparrow 4.40}$ | $92.70^{\uparrow 2.55}$ | $96.38^{\uparrow 2.18}$ | $89.45^{\uparrow 7.63}$ | $79.35^{\uparrow 1.45}$ | $90.94^{\uparrow 4.70}$ |
| Acc. (CTR-GCN) (%) | 72.73 | 86.45 | 53.68 | 79.49 | 91.24 | 93.12 | 87.64 | 76.09 | 90.94 |
| Acc. (CTR-GCN w/ MMCL) (%) | $74.91^{\uparrow 2.18}$ | $88.28^{\uparrow 1.83}$ | $61.40^{\uparrow 7.72}$ | $87.91^{\uparrow 8.42}$ | $92.70^{\uparrow 1.46}$ | $97.10^{\uparrow 3.98}$ | $89.45^{\uparrow 1.81}$ | $78.62^{\uparrow 2.53}$ | $91.67^{\uparrow 0.73}$ |

| Action Label | 34 | 38 | 47 | 48 | 67 | 68 | 72 | 73 | 76 |
|---|---|---|---|---|---|---|---|---|---|
| Acc. (CTRS-GCN) (%) | 87.68 | 90.22 | 84.42 | 84.00 | 73.82 | 80.52 | 41.39 | 32.22 | 55.50 |
| Acc. (CTRS-GCN w/ MMCL) (%) | $88.40^{\uparrow 0.72}$ | $92.75^{\uparrow 2.53}$ | $86.95^{\uparrow 2.53}$ | $85.82^{\uparrow 1.82}$ | $78.36^{\uparrow 4.54}$ | $82.26^{\uparrow 1.74}$ | $41.91^{\uparrow 0.52}$ | $34.33^{\uparrow 2.11}$ | $60.38^{\uparrow 4.88}$ |
| Acc. (CTR-GCN) (%) | 90.94 | 91.30 | 86.23 | 84.36 | 78.88 | 79.83 | 40.17 | 34.50 | 63.18 |
| Acc. (CTR-GCN w/ MMCL) (%) | $93.12^{\uparrow 2.18}$ | $94.93^{\uparrow 3.63}$ | $89.13^{\uparrow 2.90}$ | $88.00^{\uparrow 3.64}$ | $79.06^{\uparrow 0.18}$ | $84.00^{\uparrow 4.17}$ | $47.65^{\uparrow 7.48}$ | $39.75^{\uparrow 5.25}$ | $67.36^{\uparrow 4.18}$ |

| Action Label | 78 | 79 | 82 | 89 | 93 | 100 | 107 | 110 | 116 |
|---|---|---|---|---|---|---|---|---|---|
| Acc. (CTRS-GCN) (%) | 65.27 | 76.00 | 68.17 | 78.43 | 80.38 | 93.21 | 68.75 | 76.17 | 93.06 |
| Acc. (CTRS-GCN w/ MMCL) (%) | $71.90^{\uparrow 6.63}$ | $82.09^{\uparrow 6.09}$ | $72.35^{\uparrow 4.18}$ | $80.52^{\uparrow 2.09}$ | $81.42^{\uparrow 1.04}$ | $95.82^{\uparrow 2.61}$ | $70.49^{\uparrow 1.74}$ | $81.39^{\uparrow 5.22}$ | $97.05^{\uparrow 3.99}$ |
| Acc. (CTR-GCN) (%) | 74.35 | 76.52 | 69.91 | 81.21 | 80.03 | 94.95 | 71.35 | 70.78 | 91.67 |
| Acc. (CTR-GCN w/ MMCL) (%) | $76.27^{\uparrow 1.92}$ | $78.96^{\uparrow 2.44}$ | $71.48^{\uparrow 1.57}$ | $81.74^{\uparrow 0.53}$ | $82.81^{\uparrow 2.78}$ | $95.64^{\uparrow 0.69}$ | $75.52^{\uparrow 4.17}$ | $79.48^{\uparrow 8.70}$ | $94.44^{\uparrow 2.77}$ |

**Table 2: Accuracy comparison with state-of-the-art methods. ✓means use multi-stream ensemble (e.g. joint+bone).**

| Modality | | Method | Source | NTU 60 (%) | | NTU 120(%) | | NW-UCLA(%) |
|---|---|---|---|---|---|---|---|---|
| Training | Inference | | | X-Sub | X-View | X-Sub | X-Set | |
| Pose | Pose | Shift-GCN (✓) [8] | CVPR'20 | 90.7 | 96.5 | 85.9 | 87.6 | 94.6 |
| Pose | Pose | DynamicGCN (✓) [60] | ACMMM'20 | 91.5 | 96.0 | 87.3 | 88.6 | - |
| Pose | Pose | MS-G3D (✓) [38] | CVPR'20 | 91.5 | 96.2 | 86.9 | 88.4 | - |
| Pose | Pose | MST-GCN (✓) [7] | AAAI'21 | 91.5 | 96.6 | 87.5 | 88.8 | - |
| Pose | Pose | MG-GCN (✓) [5] | ACMMM'21 | 92.0 | 96.6 | 88.2 | 89.3 | - |
| Pose | Pose | CTR-GCN (✓) [6] | ICCV'21 | 92.4 | 96.8 | 88.9 | 90.6 | 96.5 |
| Pose | Pose | PSUMNet (✓) [39] | ECCV'22 | 92.9 | 96.7 | 89.4 | 90.6 | - |
| Pose | Pose | ACFL-CTR (✓) [51] | ACMMM'22 | 92.5 | 97.1 | 89.7 | 90.9 | - |
| Pose | Pose | SAP-CTR (✓) [17] | ACMMM'22 | 93.0 | 96.8 | 89.5 | 91.1 | - |
| Pose | Pose | InfoGCN (✓) [9] | CVPR'22 | 93.0 | 97.1 | 89.8 | 91.2 | 97.0 |
| Pose | Pose | FR-Head (✓) [64] | CVPR'23 | 92.8 | 96.8 | 89.5 | 90.9 | 96.8 |
| Pose | Pose | SkeletonGCL (✓) [19] | ICLR'23 | 93.1 | 97.0 | 89.5 | 91.0 | 96.8 |
| Pose | Pose | Koopman (✓) [52] | CVPR'23 | 92.9 | 96.8 | 90.0 | 91.3 | 97.0 |
| Multi-modality | Multi-modality | VPN (✓) [10] | ECCV'20 | 93.5 | 96.2 | 86.3 | 87.8 | 93.5 |
| Multi-modality | Multi-modality | TSMF (✓) [3] | AAAI'21 | 92.5 | 97.4 | 87.0 | 89.1 | - |
| Multi-modality | Multi-modality | DRDIS (✓) [54] | TCSVT'22 | 91.1 | 94.3 | 81.3 | 83.4 | - |
| Multi-modality | Pose | LST (✓) [56] | ICCV'23 | 92.9 | 97.0 | 89.9 | 91.1 | 97.2 |
| Multi-modality | **Pose** | **MMCL (Ours)** | - | **93.5** | **97.4** | **90.3** | **91.7** | **97.5** |

alignment with the global skeleton features, the RGB features extracted by CNN will go through the MLP operation $\mathcal{M}_{align}$. This alignment operation facilitates subsequent contrastive learning between two features.

Contrastive learning is popular and widely used in deep learning, especially playing a key role in multimodal learning, which aims to learn the common features between similar instances and helps to distinguish similar human actions. Our MMCL conducts contrastive learning based on the framework [68] between the RGB features $\mathcal{F}_c^i$ output by FAM and the skeleton features $\mathcal{F}_g^i$ extracted by GCN backbone, while calculating the contrastive loss by:

$$\mathcal{L}_C = \frac{1}{2N} \sum_{i=1}^{N} [\mathcal{L}(\mathcal{F}_g^i, \mathcal{F}_c^i) + \mathcal{L}(\mathcal{F}_c^i, \mathcal{F}_g^i)], \quad (4)$$

$$\mathcal{L}(\mathcal{F}_g^i, \mathcal{F}_c^i) = log \frac{e^{\theta(F_g^i, F_c^i)/\tau}}{e^{\theta(F_g^i, F_c^i)/\tau} + Neg(\mathcal{F}_g^i, \mathcal{F}_c^i)}, \quad (5)$$

$$Neg(\mathcal{F}_g^i, \mathcal{F}_c^i) = \sum_{k \neq i} (e^{\theta(F_g^i, F_c^k)/\tau} + e^{\theta(F_g^i, F_g^k)/\tau}), \quad (6)$$

where $\theta$ is the cosine similarity and $\tau$ is a temperature parameter. Note that the skeleton features extracted by the GCN backbone will first go through a MeanPooling layer to obtain the global features. Since the purpose of contrastive learning is to learn common features between similar instances, and the common features between RGB videos and skeletons of the same action are the motion dynamics of this action, so the FAM can promote the learning of motion dynamics from skeleton.

*3.3.2 Feature Refinement Module.* Since LLMs have powerful text understanding and generation capabilities, they have been used in auxiliary training for many vision tasks successfully. Multimodal LLMs that receive multimodal input are able to understand the content of multiple modalities, thereby generating more robust and comprehensive text features. In Fig. 4(b), we demonstrate the generative capabilities of different multimodal LLMs by using different RGB images and text instructions. In fact, the text instructions of our MMCL are established based on skeletal defects (e.g. the skeletal modality lacks object information) and can be readily modified as

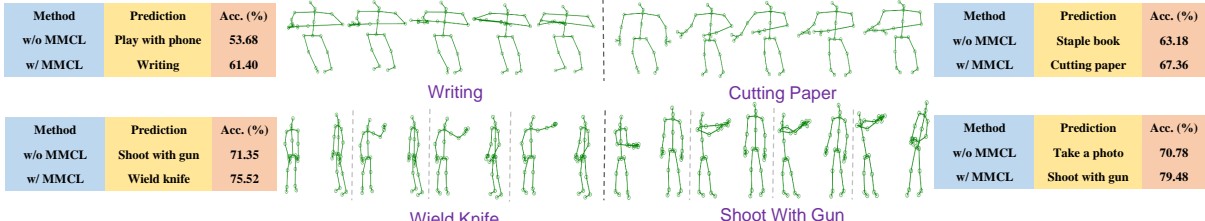

**Figure 6: Visualization of improved accuracy about difficult action samples when CTR-GCN used MMCL. The second column represents the prediction of models for the currently visualized sample and the third column represents the accuracy for all samples within the currently visualized category. We selected four difficult action samples that are prone to prediction errors in CTR-GCN, which all belong to categories involving objects or are highly relevant to hands and is difficult to distinguish objects from skeleton diagrams. Our MMCL set text instructions based on skeletal defects and generated instructive features through LLMs to guide the model to focus on the modeling of human hands and objects, thus leading to significant accuracy improvements in these difficult action samples.**

**Table 3: Comparison in the case of whether to use the MMCL.**

| Methods | Modality | Acc. (%) |
|---|---|---|
| CTRS-GCN w/o MMCL | joint | 83.88 |
| **CTRS-GCN w/ MMCL** | **joint** | **84.84$^{\uparrow 0.96}$** |
| CTR-GCN w/o MMCL | joint | 85.01 |
| **CTR-GCN w/ MMCL** | **joint** | **85.79$^{\uparrow 0.78}$** |
| CTR-GCN w/o MMCL | bone | 86.34 |
| **CTR-GCN w/ MMCL** | **bone** | **87.32$^{\uparrow 0.98}$** |
| CTR-GCN w/o MMCL | joint motion | 81.23 |
| **CTR-GCN w/ MMCL** | **joint motion** | **83.19$^{\uparrow 1.96}$** |
| CTR-GCN w/o MMCL | bone motion | 81.66 |
| **CTR-GCN w/ MMCL** | **bone motion** | **82.92$^{\uparrow 1.26}$** |

per actual requirements. The FRM of our MMCL is based on the multimodal LLMs to obtain instructive text features and refine the classification scores output by a fully connected layer. In detail, the FRM inputs text instruction and RGB image into the LLMs in the form of Visual Question Answering(VQA) in Eq. 7. Then the token features of the multimodal LLM decoder will be retained.

$$\mathcal{F}_{text} = \phi(VQA(I, Q)). \tag{7}$$

$$S_R = \sum_{i=1}^{n} \mathcal{F}_{text}^i M_i S_{\mathcal{M}}. \tag{8}$$

In terms of implementation, the FRM unifies text features obtained from different input samples into the same dimension $n$ through $\phi$. Meanwhile, $n$ learnable matrices $M$ are multiplied with text features $\mathcal{F}_{text} \in R^n$ in each feature channel $i$ and optionally refine the classification scores $S_{\mathcal{M}}$. These refined scores $S_{\mathcal{R}}$ in Eq. 8 will enhance the model's robustness and generalization in a manner similar to soft labels. Intuition suggests that introducing more detailed text instructions would result in broader and more comprehensive action descriptions, which may be more helpful to the action recognition task. However, inputting complex text instructions into the MiniGPT-4 and BLIP models tends to generate more irrelevant content, which can introduce noise features to some extent. Therefore, our MMCL adopts a brief and direct text instruction based on skeletal defects, aiming to demonstrate the effectiveness of this text prompt and our MMCL framework.

In our FRM, we focus on modeling the objects that human holds since the skeleton cannot recognize objects and selectively enhance and refine the classification scores for samples with or without objects. In fact, this focus can be easily transferred to different situations and adapted to different LLMs and text instructions by adjusting the learnable matrices during the training stage. Our FRM effectively demonstrates how to transfer multimodal LLMs to action recognition in a novel and general way. In Eq. 9, the refined classification score $S_R$ and the true label $\mathcal{Y}$ of action samples will be used to calculate the refinement loss by:

$$\mathcal{L}_R = -\sum_{i}^{N} \mathcal{Y}^i log S_R^i, \tag{9}$$

where $N$ is the number of samples in a batch and $\mathcal{Y}^i$ is the one-hot presentation of the true label about action sample $i$. Actually, the FRM of our MMCL trains the network by refining the classification scores to generate pseudo-labels and use text features to supervise the model, which was also shown to be effective in [31].

### 3.4 Loss Function

In our proposed MMCL, we use Cross-Entropy (CE) loss as the classification loss. Our MMCL uses contrastive loss $\mathcal{L}_C$ and refinement loss $\mathcal{L}_R$ as the auxiliary loss to constrain the training of the whole network, thereby introducing multimodal features into skeleton-based action recognition. The complete training loss function is defined as:

$$\mathcal{L}_{loss} = \mathcal{L}_{cls} + \lambda_1 \mathcal{L}_C + \lambda_2 \mathcal{L}_R, \tag{10}$$

where $\lambda_1$ and $\lambda_2$ represent two hyper-parameters and $\mathcal{L}_{cls}$ is the classification loss.

### 3.5 Domain-Adaptive Action Recognition

It is easily observed that the distribution of training data derived from public datasets differs from that of testing data collected in real world or other datasets, leading to a domain gap between the training (source) and testing (target) inputs. Considering this situation, our MMCL introduces multi-modality co-learning based on multimodal LLMs to enhance the robustness and generalization of the model. When dealing with skeletons from other domain datasets or real world, our MMCL conducts data interpolation to align with

**Table 4: Comparison of different CNNs in FAM.**

| Methods | CNNs | Acc. (%) |
|---|---|---|
| CTRS-GCN w/o FAM | - | 83.88 |
| CTRS-GCN w/ FAM | ResNet34 | $84.44^{\uparrow 0.56}$ |
| **CTRS-GCN w/ FAM** | **Inception-V3** | $\mathbf{84.48^{\uparrow 0.60}}$ |
| CTR-GCN w/o FAM | - | 85.01 |
| CTR-GCN w/ FAM | ResNet34 | $85.57^{\uparrow 0.56}$ |
| **CTR-GCN w/ FAM** | **Inception-V3** | $\mathbf{85.62^{\uparrow 0.61}}$ |

**Table 5: Comparison of different LLMs in FRM.**

| Methods | LLMs | Acc. (%) |
|---|---|---|
| CTRS-GCN w/o FRM | - | 83.88 |
| CTRS-GCN w/ FRM | BLIP | $84.10^{\uparrow 0.22}$ |
| CTR-GCN w/o FRM | - | 85.01 |
| CTR-GCN w/ FRM | MiniGPT-4 | $85.42^{\uparrow 0.41}$ |
| **CTR-GCN w/ FRM** | **BLIP** | $\mathbf{85.60^{\uparrow 0.59}}$ |

**Table 6: Comparison of accuracy under different hyperparameter settings.**

| Methods | $\lambda_1$ | $\lambda_2$ | Acc. (%) |
|---|---|---|---|
| Baseline | - | - | 85.01 |
| MMCL | 0.2 | 0.1 | $85.76^{\uparrow 0.75}$ |
| MMCL | 0.3 | 0.1 | $85.54^{\uparrow 0.53}$ |
| MMCL | 0.1 | 0.1 | $85.59^{\uparrow 0.58}$ |
| **MMCL** | **0.1** | **0.2** | $\mathbf{85.79^{\uparrow 0.78}}$ |

**Table 7: Comparison in parameters and computation cost when inferring a single action sample.**

| Methods | Modality | Param. | FLOPs |
|---|---|---|---|
| CTRS-GCN [6] | Pose | 2.09M | 2.41G |
| Info-GCN [9] | Pose | 1.57M | 1.84G |
| CTR-GCN [6] | Pose | 1.44M | 1.79G |
| HD-GCN [24] | Pose | 1.65M | 1.89G |
| EPP-Net [33] | Skeleton+Parsing | 25.27M | 7.84G |
| STAR-Transformer [1] | Skeleton+RGB | 58.49M | 18.67G |
| DRDIS [54] | Depth+RGB | 171.98M | 27.92G |
| **MMCL (Ours)** | **Pose** | **1.44M** | **1.79G** |

the model's input as shown in Fig. 5, enabling seamless action recognition. Due to the effectiveness of multi-modality co-learning during the training stage, our MMCL exhibits the ability to acquire more robust and deep skeletal feature representations. As a result, it maintains stable recognition performance when confronted with skeletal inputs from other domains. Meanwhile, owing to the strong generalization of multimodal LLMs, they can still produce high-quality textual features when confronted with RGB images from diverse domains and these textual features can also refine the classification scores through the fixed FRM.

## 4 EXPERIMENTS

**Datasets**: We conduct experiments on three well-established large-scale human action recognition datasets: **NTU-RGB+D** (NTU 60), **NTU-RGB+D 120** (NTU 120) and **Northwestern-UCLA** (NW-UCLA). **NTU-RGB+D**[43] is a widely used 3D action recognition dataset containing 56,880 skeleton sequences and human action videos, which are categorized into 60 action classes and performed

**Table 8: Comparison of using different adjacency matrices.**

| Methods | Adjacency Matrix | Acc. (%) |
|---|---|---|
| CTR-GCN | Natural Connection | 88.9 |
| MMCL | Natural Connection | 89.9 |
| **MMCL** | **HD-Graph** | **90.3** |

by 40 different performers. We follow the evaluation using two benchmarks provided in the original paper [43]. **NTU-RGB+D 120**[34] is derived from the NTU-RGB+D dataset. A total of 114,480 video samples across 120 classes are performed by 106 volunteers and captured using three Kinect V2 cameras. We also follow the evaluation using two benchmarks as outlined in the original research [34]. **Northwestern-UCLA** [48] comprises 1494 video clips spanning across 10 distinct categories. Our evaluation process aligns with the protocol in [48]. To demonstrate the generalization of our MMCL in zero-shot and domain-adaptive action recognition, we also conduct experiments on the **UTD-MHAD** and **SYSU-Action** datasets. **UTD-MHAD** [4] consists of 27 different actions performed by 8 subjects. Each subject repeated the action for 4 times, resulting in 861 action sequences in total. **SYSU-Action** [18] comprises 12 distinct actions performed by 40 participants, culminating in a collection of 480 video clips.

**Implementation details**: All experiments are conducted on two Tesla V100-PCIE-32GB GPUs. We use SGD to train our model for a total number of 110 epochs with batch size 200 and we use a warm-up strategy [16] to make the training procedure more stable. The initial learning rate is set to 0.1 and reduced by a factor of 10 at 90 and 100 epochs, the weight decay is set to 4e-4. We use CTR-GCN [6] as the GCN backbone and use the InceptionV3 [45] to extract deep RGB features in FAM. We use YoloV5 [46] as the person detector for video frames and uniformly sample five frames from person frames to form the RGB image with temporal information. We extract instructive text features from RGB images and text instruction based on the BLIP [28] in FRM. We introduce four skeleton modalities in [6] and the six-way ensemble in [24] to achieve multi-stream fusion. In concrete implementation, we utilize the HD-Graph [24] to replace the natural connectivity of human joints, thereby altering the initialization of symbol $A$ in Eq. 1.

### 4.1 Comparison with State-of-the-Art Methods

Notably, our MMCL is the first to introduce multimodel features based on multimodel LLMs for multi-modality co-learning into skeleton-based action recognition, which achieves better performance by assisting previous GCN backbones and can be easily migrated to more advanced backbone networks. In Table 1, we demonstrate the accuracy improvement for some action categories when applying the proposed MMCL in different GCN backbones on the NTU120 X-Sub benchmark. We also compare our MMCL with the state-of-the-art methods on the NTU 60, NTU 120 and NW-UCLA datasets in Table 2. On three datasets, our model outperforms all existing methods under nearly all evaluation benchmarks.

In Table 2, on the X-Sub benchmark of the NTU 120 dataset, our MMCL using the same ensemble outperforms the baseline CTR-GCN [6], which shows that introducing multimodal features based on MMCL into the backbone will achieve better performance. On the X-Set benchmark of the NTU 120 dataset, compared with VPN

[10] and TSMF [3] that use both the skeleton and RGB modalities at the inference stage, our MAL performs better in both recognition accuracy and inference cost by only using the skeleton at the inference stage. Compared with the method [56] that use simple text modalities based on action labels and unimodal LLM, our MMCL achieved better recognition accuracy on three datasets.

## 4.2 Ablation Studies

*4.2.1 Effectiveness in different GCN backbones.* In this section, we conduct ablation experiments on the X-Sub benchmark of NTU 120 dataset by applying MMCL to the strong CTR-baseline (CTRS-GCN) and the CTR-GCN. In Table 3, we investigate the effectiveness of MMCL across different skeleton modalities and GCN backbones. By introducing the MMCL into CTR-GCN, an accuracy improvement of 0.78% and 0.98% is achieved on the joint and bone modalities respectively. The results in Table 3 show that our MMCL leads to improvements across different skeleton modalities and backbones. Besides, our MMCL can help the backbones improve the recognition accuracy of hard actions as shown in Fig. 6.

*4.2.2 Effectiveness of FAM and FRM.* In Table 4, we demonstrate the effectiveness of the FAM and show its robustness by using different CNNs. When the FAM is applied to CTRS-GCN and CTR-GCN, the recognition performance is improved. We argue that the MMCL enhances the modeling of skeleton features by performing contrastive learning between the RGB features extracted by the FAM and the global skeleton features output by the GCNs. In Table 5, we also show the effectiveness of the FRM and its robustness by using different multimodel LLMs. When the FRM based on different LLMs is applied to the baseline network, the recognition performance is improved. An interesting observation here is that the performance of FRM based on MiniGPT-4 is worse than those based on Blip. We speculate that when faced with the same text instructions, MiniGPT-4 tends to focus on describing a lot of clothing-related information and the background (e.g. the color of clothes), which inadvertently introduces unnecessary textual noise.

*4.2.3 Efficiency and robustness of MMCL.* The Table 6 present the accuracy of our MMCL under different hyperparameter settings. In Table 7, we also show the parameters and computation cost required by MMCL for inference when using a single action sample. In Table 8, we compared the recognition accuracy using different adjacency matrix on the NTU120 X-Sub benchmark. Here, replacing the naturally connected adjacency matrix with HD-Graph resulted in higher accuracy. In Table 9, we explore the recognition accuracy when different text instructions are employed based on skeletal modality defects. The results in Table 9 suggest that the FRM based on different text instructions improves the recognition performance of the baseline model. Simultaneously, they also demonstrate that our FRM can be transferred to different reasonable text instructions as needed and can generate instructive textual features through multimodal LLMs to aid in the modeling of the baseline model.

## 4.3 Model Generalization

In Table 10, we utilized a subset of UTD-MHAD dataset [4] and SYSU-ACTION dataset [18] to investigate the generalization of

**Table 9: Comparison of different text instructions in the Feature Refinement Module on NTU 120 X-Sub benchmark.**

| Methods | LLMs | Text Instruction | Acc. (%) |
|---|---|---|---|
| CTR-GCN w/o FRM | - | - | 85.01 |
| CTR-GCN w/ FRM | BLIP | Whether the hands are touching head? | $85.52^{\uparrow 0.51}$ |
| CTR-GCN w/ FRM | BLIP | Whether the hands are touching foot? | $85.44^{\uparrow 0.43}$ |
| CTR-GCN w/ FRM | BLIP | Whether the person are holding object? | $85.60^{\uparrow 0.59}$ |

**Table 10: Exploration about the generalization of MMCL and LLMs in different domains. The J represents the use of model weights that are trained on the joint modality.**

| Methods | Top-1/Top-5 Acc. (%) | |
|---|---|---|
| | UTD-MHAD | SYSU-Action |
| CTRS-GCN (J) [6] | 37.70/69.63 | 37.50/53.33 |
| CTR-GCN (J) [6] | 48.17/74.87 | 27.50/51.67 |
| HD-GCN (J CoM-1) [24] | 46.07/74.87 | 26.27/58.33 |
| **Ours (w/o BLIP Refine)** | **52.88/81.16** | **39.17/76.67** |
| **Ours (w/ BLIP Refine)** | **54.97/84.29** | **42.50/80.83** |

MMCL and LLMs. We utilize weights trained on the joint modality of NTU 120 X-Sub benchmark for zero-shot action recognition and use interpolation to extend the joints to fit the models' input as shown in Fig. 5. In zero-shot recognition, all samples from two datasets have never been involved in training and they directly use a previously well-trained model for action recognition. Therefore, the accuracy of zero-shot recognition can effectively reflect the model's generalization and transferability. In Table 10, when the backbone CTR-GCN is trained with MMCL, the top-1 and top-5 accuracy for actions from different domains are 39.17% and 76.67%, while the baseline CTR-GCN trained without MMCL are 27.50% and 51.67% in SYSU dataset. Compared to the baseline CTR-GCN and HD-GCN, our MMCL shows a positive improvement in both Top-1 and Top-5 accuracy. We believe that MMCL benefits from the RGB modality and multimodal LLMs, thus demonstrating commendable generalization and robustness when facing other datasets in different domains. Besides, when introducing text features generated by multimodal LLMs to refine the zero-shot scores based on the Eq. 8, we are surprised to find an improvement in both recognition accuracy. This indicates that soft labels based on multimodal LLMs can be transferred to the domain adaptive action recognition. We argue that multimodal LLMs exhibit strong generalization capabilities, as they also can generate robust and effective text features when presented with RGB images in different domains.

## 5 CONCLUSION

We present a novel Multi-Modality Co-Learning (MMCL) framework for efficient skeleton-based action recognition, which is the first to introduce multimodal features based on multimodal LLMs for multi-modality co-learning into action recognition. The MMCL guides the modeling of skeleton features through complementary multimodal features and maintains the network's simplicity by using only skeleton in inference. Meanwhile, our MMCL is orthogonal to the backbone networks and thus can be coupled with various GCN backbones and multimodal LLMs. The effectiveness of our proposed MMCL is verified on three benchmark datasets, where it outperforms state-of-the-art methods.

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
