# OpenReview forum: "Multi-Modality Co-Learning for Efficient Skeleton-based Action Recognition"
_acmmm.org/ACMMM/2024/Conference — MM2024 Poster_

### Official Review · Reviewer_VTck · 2024-05-19

**Rating:** 5
**Confidence:** 4

**Summary:**

This paper presents multi-modality co-learning (MMCL) framework for skeleton-based action recognition by adopting additional RGB video and text modalities. During training, RGB features are aligned with skeleton features to facilitate contrastive learning.

**Strengths:**

The paper is well-written.

The proposed method achieves the best results on NTU 60, NTU 120, and NW-UCLA datasets, albeit not significantly higher than SkeletonGCL or Koopman.

**Limitations:**

Using RGB modality to assist in the learning of skeleton modality can be considered somewhat innovative, but the significant difference between RGB modality and skeleton modality may mean that simply extracting features and aligning them may not improve the experimental results.

The author's claim of Multi-Modality Co-Learning simply involves stacking RGB videos, text, and skeleton modalities, extracting features and aligning them in a straightforward manner. The method lacks significant highlights and is not implemented in a sufficiently detailed manner.

While there are no grammatical errors in the writing, the writting can be further improved, for example, placing experimental results in the Method section.

**Suitability:**

2

---

### Official Review · Reviewer_SZMk · 2024-05-24

**Rating:** 3
**Confidence:** 3

**Summary:**

This study introduces MMCL, a framework for skeleton-based action recognition. The proposed MMCL uses large language models (LLMs) as auxiliary networks during training. This co-learning approach leverages both skeleton and multimodal data. MMCL relies solely on skeleton data. The framework consists of two modules: one that aligns visual features from video frames with skeleton features, and another that uses LLMs to generate instructive text features from RGB images, temporal information, and text instructions. These features refine classification scores, enhancing the model's robustness and generalization. Experiments are conducted on NTU RGB+D, NTU RGB+D 120 and Northwestern-UCLA benchmarks, also the UTD-MHAD and SYSU-Action datasets for zero-shot and adaptive action recognition.

**Strengths:**

The paper explores the multimodal contrastive learning methods for the research on skeleton-based action recognition. .

The experiments are conducted with multiple datasets, NTU RGB+D, NTU RGB+D 120 and Northwestern-UCLA, also the UTD-MHAD and SYSU-Action datasets.

The experiment part showcases sufficient ablation study and analysis of the proposed methods in different perspectives.

**Limitations:**

The contrastive learning method employed in FAM is based on the framework presented in [68].  However, the paper lacks a clear explanation of the differences between the proposed method and [68].  A detailed description of the technical contributions specific to FAM is necessary to understand the novelty of the contrastive learning approach within this context.

The FRM relies on visual question answering (VQA) as the text-based task. Exploring alternative approaches, such as video captioning, referring, or summarization, maybe improve this module. Additionally, investigating the use of multi-tasking for feature alignment, rather than relying solely on VQA, is a possible plus.

The paper focuses on using Graph Convolutional Networks (GCN) as the backbone architecture.  While this is a reasonable choice, hwo about exploring the performance and effectiveness of applying the proposed methods (FAM and FRM) to other backbone types, particularly transformer-based architectures. This would provide insights into the generalizability of the approach.

Table 2 shows that MMCL achieves only marginal improvements compared to existing methods that use multi-modal data during training and pose data during inference. This raises concerns about the overall effectiveness of the co-learning approach.

A clear baseline for the proposed model is still missing. Table 2 lacks information on the results achievable by using both pose data for training and inference. Establishing this baseline is essential for evaluating the true impact of approach.

The paper does not explore the performance of MMCL when using multi-modal data during the inference stage. Understanding how the model behaves when incorporating both modalities throughout the process would provide valuable insights into its overall efficiency and effectiveness, particularly regarding potential improvements or trade-offs compared to the proposed co-learning approach.

**Suitability:**

3

---

### Official Review · Reviewer_sX4M · 2024-05-25

**Rating:** 4
**Confidence:** 4

**Summary:**

This paper proposes Multi-Modality Co-Learning (MMCL) for skeleton-based action recognition . It leverages a Feature Alignment Module (FAM) aligning RGB video features with skeleton data through contrastive learning, and a Feature Refinement Module (FRM) utilizing RGB images and textual instructions to refine classification scores.

**Strengths:**

1. The paper is clear and easy to understand.
2. The figures in the paper are well-designed and aid in comprehension.
3. The experiments are comprehensive, and the results demonstrate the effectiveness of the proposed method.

**Limitations:**

1. While it is acknowledged that multimodal learning is a recent trend, the reliance on dataset creation and acquisition for RGB video modality in skeleton-based action recognition is not sufficiently addressed. Considering recent generative methods such as diffusion, which can effectively expand skeleton data. This limitation affects the scalability of the method for skeleton-based action recognition tasks.
2. The main components of the MMCL model seem to rely on existing methods, lacking innovation. For instance, the core innovation of Multi-Modality Co-Learning, primarily achieved through contrastive learning, the design draws heavily from the approach outlined in reference [56] LST.
3. Figure 2 does not effectively showcase the superiority of the model. Since the eight categories depicted in Figure 2 are already selected as the top-performing results, it doesn't provide necessary insights into the model's performance.

**Suitability:**

3

---

### Official Review · Reviewer_bNKM · 2024-05-26

**Rating:** 4
**Confidence:** 3

**Summary:**

This paper introduce a novel framework named multi-modality co-learning(MMCL) to achieve efficient skeleton-based action recognition. By incorporating LLMs to generate textual descriptions of images for co-learning in training, this method empowers the mainstream GCN module to produce robust feature representations. Only skeletons GCN module is used for inference due to the effective co-learning. Extensive experiments shows the effectiveness of the MMCL framework and its efficacy.

**Strengths:**

1. The MMCL framework is the first to introduce multi modal LLMs in skeleton-based action recognition.
2. The experiments are sufficient and have demonstrated the effectiveness of the proposed method.

**Limitations:**

1. The writing of methodology needs to be improved. For example, the VQA module and  in eq.7 should be introduced in detail. And the formula of  in eq.10 is not mentioned.
2. The sec 3.5 mentions the MCCL have the ability of domain-adaptive, but there lacks of corresponding experiments to prove it.
3. Lacks of qualitative results to prove the effectiveness of multi-modality co-learning.

**Suitability:**

3

---

### Meta-Review · Area_Chair_zofw · 2024-07-02

**Recommendation:** Accept (Poster)
**Confidence:** 5

**Metareview:**

Pros:
- The paper is the first to introduce multi modal LLMs in skeleton-based action recognition.
- The experiments are sufficient and have demonstrated the effectiveness of the proposed method.
- The paper is well written and easy to understand.

Cons:
- The contrastive learning method employed in this paper lacks novelty.
- The performance improvement is marginal